# Retinal transplantation of photoreceptors results in donor–host cytoplasmic exchange

Tiago Santos-Ferreira[1],*, Sílvia Llonch[1],*, Oliver Borsch[1],*, Kai Postel[1], Jochen Haas[1] & Marius Ader[1]

Pre-clinical studies provided evidence for successful photoreceptor cell replacement therapy. Migration and integration of donor photoreceptors into the retina has been proposed as the underlying mechanism for restored visual function. Here we reveal that donor photoreceptors do not structurally integrate into the retinal tissue but instead reside between the photo-receptor layer and the retinal pigment epithelium, the so-called sub-retinal space, and exchange intracellular material with host photoreceptors. By combining single-cell analysis, Cre/lox technology and independent labelling of the cytoplasm and nucleus, we reliably track allogeneic transplants demonstrating cellular content transfer between graft and host photoreceptors without nuclear translocation. Our results contradict the common view that transplanted photoreceptors migrate and integrate into the photoreceptor layer of recipients and therefore imply a re-interpretation of previous photoreceptor transplantation studies. Furthermore, the observed interaction of donor with host photoreceptors may represent an unexpected mechanism for the treatment of blinding diseases in future cell therapy approaches.

[1] CRTD/Center for Regenerative Therapies Dresden, Technische Universität Dresden, Fetscherstrasse 105, 01307 Dresden, Germany. * These authors contributed equally to the work. Correspondence and requests for materials should be addressed to M.A. (email: marius.ader@crt-dresden.de).

Photoreceptor transplantation has been shown to repair retinal function in mouse models of retinal degeneration[1–3]. Therefore, it is considered as a potential cell replacement treatment option for retinopathies characterized by photoreceptor loss such as retinitis pigmentosa and age-related macular degeneration. Following sub-retinal, that is, between the photoreceptor/outer nuclear layer (ONL) and retinal pigment epithelium, transplantation of photoreceptor precursors into partially degenerated adult retinas, several reports provided evidence for successful migration, integration and maturation of donor cells into the host's ONL[1–6].

In most transplantation studies, donor cells expressing a fluorescent reporter are used to facilitate their identification within the host tissue. However, sole reporter detection can be misleading, as it has also been found in host cells due to donor–host cell fusion[7–9]. Indeed, transplantation studies performed more than a decade ago using Cre/LoxP technology and detection of fluorescent bi-nucleated heterokaryons demonstrated spontaneous fusion between bone marrow-derived donor cells and diverse host cell types, including Purkinje neurons, cardiomyocytes and hepatocytes[9], thereby challenging the hypothesis of somatic (stem) cell transdifferentiation into multiple cell types[10].

In previous photoreceptor transplantation studies, a potential fusion between donor and host cells was ruled out, as fluorescent reporter positive photoreceptors found within the ONL contained just a single nucleus and, moreover, some fluorescently labelled processes of donor cells showed no co-labelling when grafted into hosts expressing a different fluorescent marker[4,5]. In general, exchange of cytoplasmic content between cells is not restricted to the generation of syn- or heterokaryons, that is, the generation of a single cell containing two nuclei, but can also be observed between morphologically distinct cells that form membrane connections[10] including plasma bridges/tunnelling nanotubes[11,12] or transfer intracellular material by vesicular transport processes[13].

In this study, we reinvestigated the potential occurrence of fusion events following allogeneic photoreceptor transplantation by (i) single-cell analysis using flow cytometry, (ii) the Cre/LoxP fusion assay and (iii) separating cytoplasmic from nuclear labelling. With these experiments we observe that, after retinal transplantation, the majority of grafted photoreceptors do not structurally integrate into recipient tissue but instead exchange intracellular material with host photoreceptors.

## Results

**Double fluorescent photoreceptors after transplantation.** Initially, postnatal day (P) 4 photoreceptors, isolated from reporter mice expressing cytoplasmic enhanced green fluorescent protein (eGFP) driven by the rod-specific neural retina leucine zipper promoter (Nrl-eGFP[14]), were transplanted into the sub-retinal space of recipient mice ubiquitously expressing DsRED[15], to assess whether single cells within host retinas contained both reporters (Fig. 1a). Three weeks after grafting, flow cytometric analysis identified cell populations within host retinas expressing eGFP or DsRED, as well as a robust number of double-labelled (eGFP$^+$/DsRED$^+$) cells ($98 \pm 17$ cells per 1 million photoreceptors; $n = 3$; Fig. 1b). Further analysis of individual experimental retinas ($n = 3$) by imaging flow cytometry, a technique allowing concurrent visualization of individual events during flow cytometry (Supplementary Fig. 1), revealed that 80% of double-positive cells indeed expressed both fluorescent reporters, that is, GFP and DsRED (Fig. 1c). In contrast, in control samples containing a mixture of enriched P4 GFP$^+$ photoreceptors with DsRED$^+$ retinal cells, double positivity was only observed due to

cell doublets or attached fluorescent debris (Fig. 1c). These results suggest the exchange of cytoplasmic content or fusion between donor and host photoreceptors.

**Exchange of intracellular content between donor and host.** To directly assess the exchange of intracellular content between donor and host cells, photoreceptors isolated from a conditional mouse reporter line containing a loxP-flanked stop cassette upstream of the *tdTomato* reporter gene (Ai9 mice[16]) were transplanted into photoreceptor-specific Cre-recombinase expressing hosts (B2-Cre$^{+/-}$; $n = 6$; Fig. 2a)[17]. Three weeks after grafting, donor cells located in the sub-retinal space ($357 \pm 96$ cells per retina; Fig. 2f) as well as photoreceptors within the host ONL ($443 \pm 103$ cells per retina; Fig. 2f) showed expression of tdTomato (Fig. 2b–e), whereas no reporter expression was detected on transplantation of Ai9 donor cells into B2-Cre$^{-/-}$ recipients (Fig. 2f and Supplementary Fig. 2). This result indicates that Cre-recombinase produced by host photoreceptors translocates into donor cells enabling the excision of the stop cassette and thus expression of the reporter protein. The detection of the fluorescent reporter also within photoreceptors of the recipient's ONL suggests a bidirectional exchange of intracellular material between donor and host cells or, alternatively, could be due to integration of fluorescent-labelled cells, following unidirectional transfer of Cre-recombinase, into the ONL.

**GFP$^+$ cells in the host ONL lack a donor-derived nucleus.** To distinguish between potential integration of donor photoreceptors and fusion events, donor cells with the nucleus and cytoplasm independently labelled were used for grafting experiments. Therefore, Nrl-eGFP mice were repeatedly injected during development with the thymidine analogue 5-ethynyl-2-deoxyuridine (EdU) to label nuclei of proliferating cells within the retina (Supplementary Fig. 3a–d). Subsequently, P4 photoreceptor precursors containing cytoplasmic GFP and EdU-labelled nuclei were sub-retinally transplanted into adult wild-type mice. Three weeks after transplantation, the ONL of experimental retinas ($n = 3$) contained GFP-labelled cells with mature photoreceptor morphology (Supplementary Fig. 3e) located next to clusters of GFP$^+$ grafts residing in the sub-retinal space. Although the majority of GFP$^+$ cells placed in the sub-retinal space were also EdU$^+$ (Supplementary Fig. 3e–h), no EdU labelling was observed in GFP$^+$ cells located within the ONL (Supplementary Fig. 3e,i–k). These results suggest that integration of donor cells into the host ONL does not occur or is a rare event; instead, cytoplasmic content, but not the entire cell nuclei/nuclear genomic DNA, might be exchanged from donor cells to host photoreceptors.

As consecutive EdU injections during development did not label all photoreceptors of the P4 retina (Supplementary Fig. 3b–d), Y-chromosome fluorescent *in situ* hybridization (FISH) was used to identify all nuclei within the donor cell fraction (Fig. 3 and Supplementary Fig. 4), allowing quantitative analysis. Therefore, donor photoreceptors were isolated from male Nrl-eGFP mice and transplanted into female wild-type recipients (Fig. 3a). After 3 weeks, virtually all analysed GFP$^+$ cells within the sub-retinal space contained a Y-chromosome identifying them as donor photoreceptors ($98.24 \pm 0.77\%$; Fig. 3b–f,k). In contrast, of all GFP$^+$ photoreceptors located within the host ONL, only few cells showed co-staining for the Y-chromosome ($1.2 \pm 0.42\%$; Fig. 3b,g–j,k, l–o), confirming that cytoplasmic content but not the nucleus of donor cells is translocated into host photoreceptors.

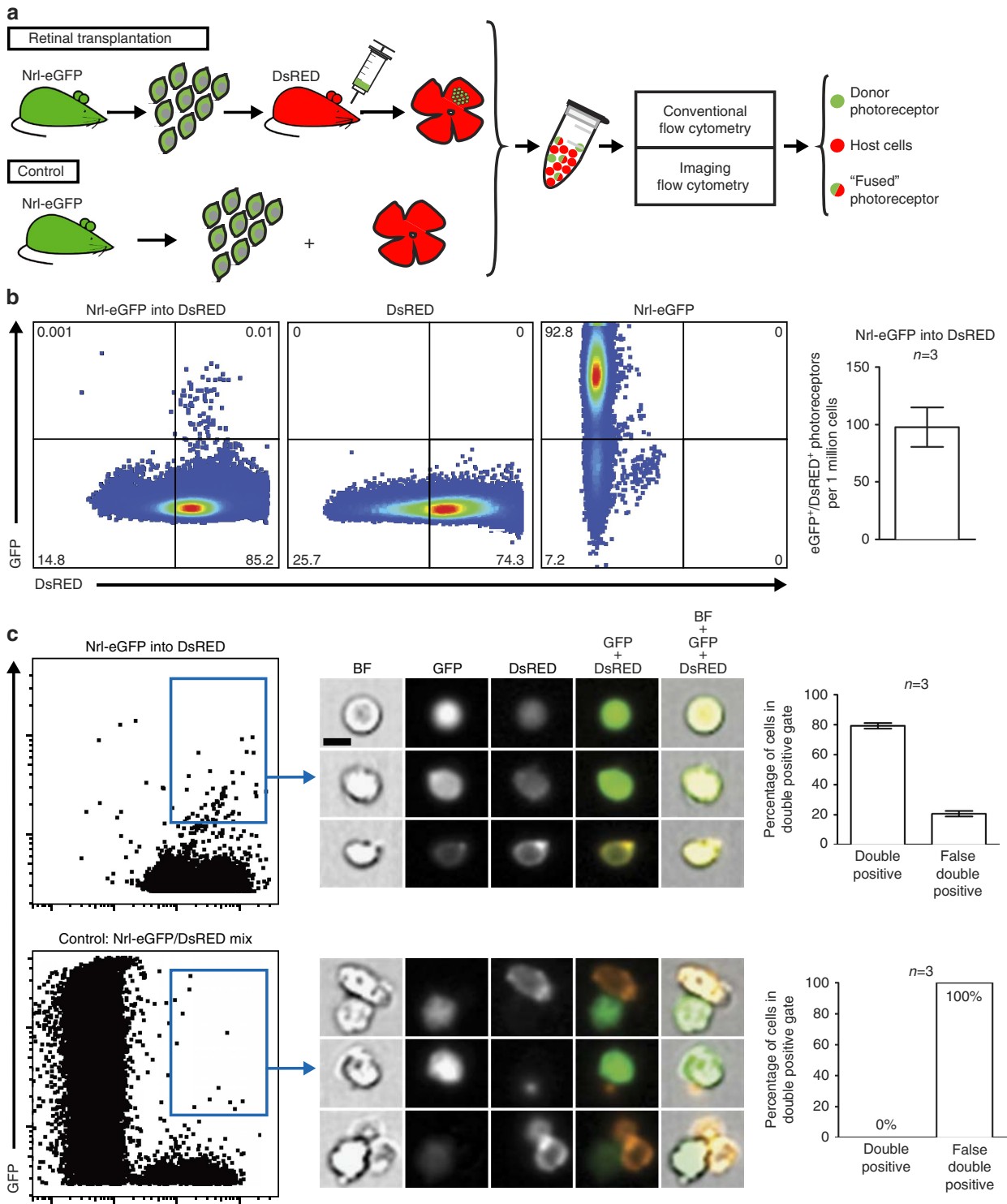

**Figure 1 | Analysis of DsRED host retinas grafted with Nrl-eGFP photoreceptors by conventional and imaging flow cytometry. (a)** Schematic representation of experimental outline: transplantation of P4 eGFP[+] photoreceptors into the retina of adult DsRED hosts and corresponding control (mixing of enriched P4 eGFP[+] photoreceptors with DsRED[+] retinal cells) with subsequent conventional and imaging flow cytometry analysis. **(b)** Flow cytometric analysis of individual experimental retinas ($n=3$) revealed a robust number of GFP[+]/DsRED[+] double-labelled photoreceptors ($98\pm17$ photoreceptors per 1 million photoreceptors) within DsRED host retinas 3 weeks after transplantation of eGFP[+] donor cells (**b**, left plot), indicating exchange of cytoplasmic material between donor and host photoreceptors. Nrl-eGFP (**b**, right plot) or DsRED (**b**, middle plot) retinas do not contain double-positive cells. **(c)** In transplanted retinas (Nrl-eGFP into DsRED; $n=3$), imaging flow cytometry identified single GFP[+]/DsRED[+] cells (three exemplary cells are shown) in the double-positive cell population (upper left plot, blue gate). The majority of cells ($\sim80\%$; upper right graph) showed double fused fluorescence, whereas $\sim20\%$ were false double-positive (that is, containing doublets or attached fluorescence debris). All double-positive events in Nrl-eGFP mixed DsRED controls (lower left plot, blue gate; lower right graph; $n=3$) were identified as false double-positive (100%; three exemplary cells are shown). Scale bar, 5 µm. Data represent mean ± s.e.m. BF, bright field.

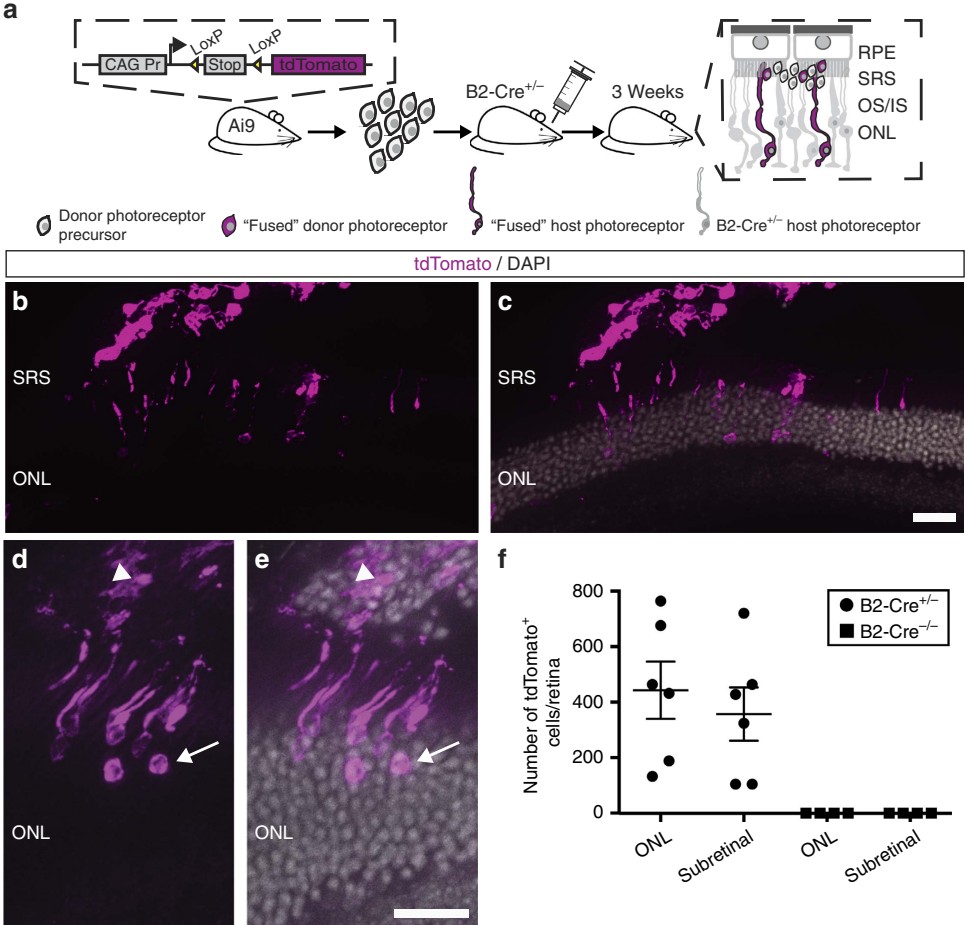

**Figure 2 | Visualization of fusion events following photoreceptor transplantation using Cre/LoxP technology. (a)** Schematic representation of experimental outline to visualize transfer of cytoplasmic material between donor and host cells by Cre/LoxP recombination. **(b–e)** Donor photoreceptors isolated from floxed reporter mice (Ai9) and transplanted into rod photoreceptor-specific Cre mice (B2-Cre[+/−]) show expression of the tdTomato reporter in sub-retinal (SRS) located cells **(b–e**; arrow head in **d,e)** revealing transfer of Cre-recombinase from host photoreceptors to donor cells. tdTomato is also present in photoreceptors located in the ONL **(b–e**; arrow in **d,e)**, suggesting bidirectional transport of intracellular content between donor and host photoreceptors. **(f)** Quantification of tdTomato[+] cells in the sub-retinal space and in the ONL of B2-Cre[+/−] hosts (n = 6) show similar amounts of reporter-labelled cells (ONL: 443 ± 103 cells per retina; SRS: 357 ± 96 cells per retina). No tdTomato-expressing cells were detected in transplanted B2-Cre[−/−] mice (n = 4). Data represent mean ± s.e.m. Scale bar,: 20 μm **(c,e)**. CAGS Pr, CAGS promoter; INL, inner nuclear layer; IS/OS, inner/outer segments; ONL, outer nuclear layer; RPE, retinal pigment epithelium; SRS, sub-retinal space.

## Discussion

Photoreceptor replacement therapies have shown promising results leading to functional restoration of rod and cone-mediated vision[1,3]. Migration and integration of grafted photoreceptors into the host's retinal circuitry has been hypothesized as the underlying mechanism for vision improvement[18]. However, our results reveal that *de novo* integration of photoreceptor transplants is a rare event. Conversely, sub-retinally located donor cells exchange cytoplasmic content with endogenous photoreceptors. Thus, potential restoration of visual function might have been achieved by material transfer, suggesting a new powerful mechanism to rescue dysfunctional and degenerating photoreceptors. Our findings will have strong implications for a re-evaluation of the field of photoreceptor transplantation.

Interestingly, reporter expression within the host retinal tissue was only observed in the ONL and exclusively detected in photoreceptors, suggesting that exchange of intracellular content is restricted to photoreceptor–photoreceptor interactions. Indeed, transplantation of other fluorescent retinal cell types, for example, retinal progenitors or CD73[−] cells, resulted only in very few reporter-labelled photoreceptors in the ONL[4,6,19]. In addition,

recent data on transplantation of photoreceptors into mouse models of severe retinal degeneration with complete loss of photoreceptors did not result in cells within the recipient's retinal tissue with reporter labelling[20,21], suggesting that retinal cells of the inner nuclear layer, that is, horizontals, bipolars, Müller glia or amacrines, do not exchange cytoplasmic content with donor photoreceptors. Notably, grafted photoreceptors in severely degenerated retinas still showed signs of a mature phenotype including expression of phototransduction proteins, outer segment-like structures and synapse formation that resulted in some functional improvements[20,21]. Therefore, disease states characterized by complete loss of photoreceptors represent a promising target for photoreceptor replacement approaches in future clinical applications.

Previous studies reported fluorophore-labelled cells within the host ONL, not only in wild-type hosts but also in several genetic degeneration mouse models that contain dysfunctional photo-receptors[1–4]. Further studies need to be performed, to evaluate whether donor–host transfer of cytoplasmic content occurs in the diseased retina as well. Interestingly, the described morphology of fluorophore-labelled cells within the ONL strongly correlated

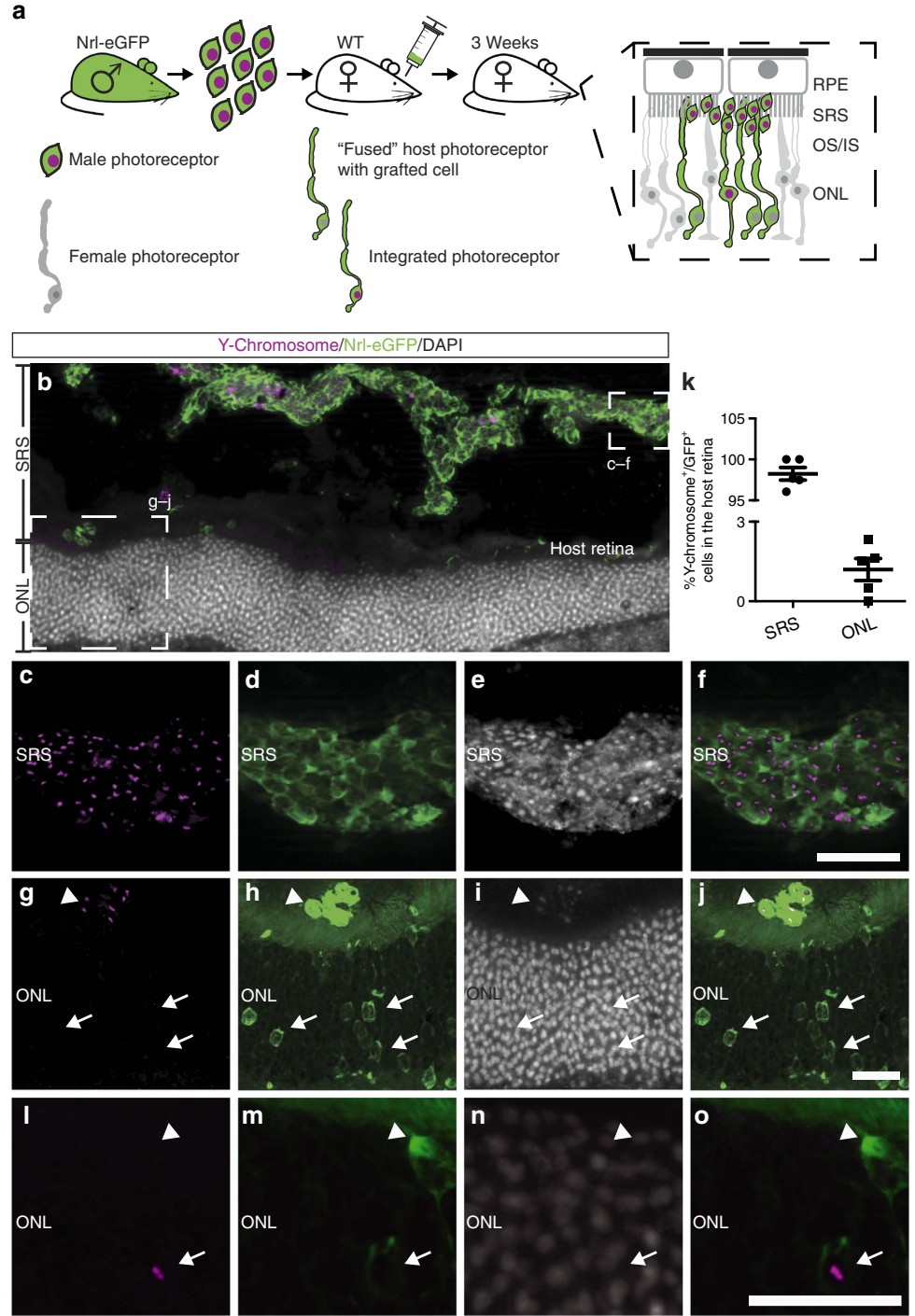

**Figure 3 | Donor cell-derived eGFP but no nuclei are located in host photoreceptors.** (**a**) Experimental overview: Nrl-eGFP male photoreceptor precursors were transplanted into wild-type female retinas. Sections of individual experimental retinas ($n = 5$) were analysed by Y-chromosome FISH and immunohistochemistry for eGFP. (**b**) Overview of grafted area 3 weeks after transplantation (increased space between graft and host ONL results from a histological (cutting) artefact). (**c–f**) Y-chromosome$^+$/GFP$^+$ cells were located in the SRS, identifying them as donor photoreceptors (see also arrowhead in **g–j**). (**g–j**) Y-chromosome$^-$/GFP$^+$ cells were detected in the ONL of the host retina (arrows), demonstrating transfer of donor cytoplasmic content but not nuclei to host photoreceptors. (**k**) Quantification of GFP$^+$ cells containing Y-chromosome in the SRS and in the ONL of the host. Although the vast majority of the GFP$^+$ cells in the SRS contained a Y-chromosome ($98.24 \pm 0.77\%$), only $1.2 \pm 0.42\%$ of GFP$^+$ cells in the ONL showed a positive staining for the Y-chromosome. Data represent mean $\pm$ s.e.m. (**l–o**) Very few GFP$^+$ cells within the ONL of the host retina were also Y-chromosome$^+$, indicating a low extent of structural integration of donor cells into the host retina. Scale bars, 20 µm (**f,i,j,o**). INL, inner nuclear layer; IS/OS, inner/outer segments; ONL, outer nuclear layer; RPE, retinal pigment epithelium; SRS, sub-retinal space.

with the morphology of host photoreceptors in different degeneration models[2], suggesting a potential fusion of donor and host photoreceptors also in a disease environment. Besides the presence of fluorescent reporters, correct expression of proteins missing in the respective disease models was observed in ONL located photoreceptors and even functional recovery was

achieved following photoreceptor transplantation in dysfunctional, mildly degenerated retinas[1–3]. Based on our findings, we conclude that the majority of donor cells remain in the sub-retinal space and thus may not form direct synaptic connections to endogenous secondary neurons. Hence, we hypothesize that the observed visual improvements[1–3] might result from remaining endogenous photoreceptors supplemented by donor cell-derived proteins. Interestingly, reporter labelled photoreceptors within the ONL have been observed up to 6 months following transplantation, although in significantly lower numbers than after 4 weeks[2,22], implying stable photoreceptor–photoreceptor connections or, alternatively, dynamic re-arrangements of cell–cell contacts over long periods of time given that GFP has a half-life of ∼26 h[23]. Improving fusion efficiency, as reported for transplanted bone marrow cells with Purkinje neurons in conditions such as inflammation, radiation exposure, chemotherapeutic drugs, age or damage[24] and increasing long-time survival of transplanted photoreceptors might therefore represent a new opportunity for cell-based treatment approaches for photoreceptor degenerative diseases. Interestingly, a recent study provided evidence that damage-induced fusion of haematopoietic stem and progenitor cells with retinal ganglion or amacrine cells contributed to retinal tissue repair[25].

In conclusion, photoreceptor transplantation into host retinas with remaining photoreceptors results in photoreceptor–photoreceptor biomaterial exchange rather than migration and integration into the recipient's ONL. Transfer of cellular content between donor and host cells has to be taken into account when interpreting experiments using photoreceptor transplantation into partially degenerated retinas with the intention to achieve cell replacement. In fact, the potential exchange of cellular material in any transplantation setting should be rigorously evaluated given the increasing number of reports providing evidence for complete[9,25,26] or partial (our study) cell–cell fusion events. Such knowledge will be an important prerequisite to determine the impact of donor cells in tissue repair[27]. Although the mechanism of vision improvement by photoreceptor transplantation may be different than originally conceptualized, capitalizing on the fusion-mediated exchange of cytoplasmic components may represent a productive direction for the development of cell support therapies.

## Methods

**Photoreceptor transplantation into the mouse retina.** Photoreceptors were isolated from postnatal day (P) 4 Nrl-eGFP[14] or Ai9 (ref. 16; Gt(ROSA) 26Sort[m9(CAG-tdTomato)Hze], Jackson Laboratory, Bar Harbor, USA) mice, enriched by CD73-based magnetic-activated cell sorting and transplanted into the sub-retinal space of adult (6–9 week) mice[28]. Therefore, eyes from donor P4 pups were enucleated, rinsed in PBS and transferred to Hank's balance salt solution (Gibco). Next, retinas were isolated, digested to a single cell suspension using papain dissociation (Worthington Biochemical Corporation) and the resulting cell suspension was incubated with rat anti-mouse CD73 antibody (0.5 mg ml$^{-1}$ stock solution diluted 1:40, BD Biosciences, ref: 550738) followed by anti-rat IgG Microbeads (1:5 of stock solution, Miltenyi, Germany, ref: 130–048–502). Cell suspensions were rinsed, spun down, ran through LS Columns attached to a magnet and CD73$^+$ cell fractions (rod photoreceptors) were concentrated by centrifugation and resuspension to a $2 \times 10^5$ cell per µl solution. Using a blunt 34 Gauge needle attached to a 10 µl Hamilton syringe, donor cells were *trans*-vitreally transplanted into the sub-retinal space of recipients, that is, wild-type, ubiquitous DsRED (actin-DsRED, (15), B6.Cg-Tg(CAG-DsRed*MST)1Nagy/j, Jackson Laboratory) or rod photoreceptor-specific B2-Cre[17] mice. All animal experiments were approved by the ethics committee of the TU Dresden and the Landesdirektion Dresden (approval number 24-9168.11-1/2008-33, 24-9168.11-1/2012-33 and 24-9168.11-1/2013-23) and performed in accordance with the regulation of the European Union, German laws (Tierschutzgesetz), the ARVO statement for the Use of Animals in Ophthalmic and Vision Research, as well as the NIH Guide for the care and use of laboratory work.

**EdU labelling of donor photoreceptors.** The thymidine analogue EdU (20 mM; Click-iT EdU Alexa Fluor 647 Imaging Kit; Invitrogen) was intraperitoneally injected (1 µl per gram body weight) into female time-mated Nrl-eGFP mice 18 and 20 days after fertilization. Delivered pups received additional intraperitoneal EdU injections (5 mM, 4 µl per gram body weight) at P1 and P3 before retinal tissue was harvested at P4 for transplantation (see above) or histological analysis. Prior immunohistochemistry (see below) EdU was detected on 20 µm-thick retinal cryosections by following the manufacturer's instructions using AlexaFluor 647 azide for visualization.

**Immunohistochemistry.** Transplanted mice were killed 3 weeks after transplantation, eyes enucleated and fixed in 4% paraformaldehyde (Merck Millipore) for 1 h at 4 °C. The posterior part of the eye (sclera, choroid, retinal pigmented epithelium and retina) was cryopreserved overnight at 4 °C in a 30% (weight/volume) sucrose solution and embedded in optimal cutting medium (OCT, NEG50, Thermo Scientific). Twenty-micrometre-thick sections were collected on star frost slides, air-dried for 1 h at room temperature (RT), hydrated with PBS for 30 min and incubated with blocking solution containing 0.3% Triton-X100 (SERVA), 1% BSA (SERVA) and 5% donkey serum for 1 h at RT. Slides were incubated with primary antibody against GFP (10 mg ml$^{-1}$ stock solution diluted 1:1,000, AbCam, ref: AB13970) overnight at 4 °C, washed 3 times for 10 min and respective Cy2-conjugated secondary antibody (1.5 mg ml$^{-1}$ stock solution diluted 1:1,000, Jackson Immunoresearch, ref: 703225155) was added for 90 min in a 4′,6-diamidino-2-phenylindole solution (DAPI, 1:20,000, Sigma). Following incubation, slides were washed three times for 10 min in PBS and mounted in AquaPolymount (Polysciences).

**Image acquisition and processing.** Fluorescent images were acquired using a structured illumination microscope (Apotome ImagerZ1, Zeiss) and processed offline using Fiji, Adobe Photoshop and Adobe Illustrator. Adobe Illustrator was used to generate illustrations and schematic representations. Mouse illustration used for schematic representations was simple mouse (c)Seans Potato Business, CC-BY-SA-3.0.

**FISH for Y-chromosome detection.** For combined chromosomal FISH (Y-chromosome FISH) and immunohistochemistry[29], experimental retinas were collected, fixed for 1 h at 4 °C with freshly prepared 4% paraformaldehyde (Merck Millipore) and cryopreserved in 30% sucrose overnight at 4 °C. Cryosections of 10 µm thickness were rehydrated with 10 mM citrate buffer and treated with $2 \times$ saline sodium citrate buffer (SSC). Following pretreatment with 50% formamide for 1 h at RT, sections were washed in PBS for 5 min and incubated with primary antibody against eGFP (10 mg ml$^{-1}$ stock solution diluted 1:800, AbCam, ref: AB13970) overnight at 4 °C, followed by incubation with secondary antibody conjugated to AlexaFluor 488 (1.5 mg ml$^{-1}$ stock solution diluted 1:1,000, Jackson Immunoresearch, ref: 103545155) for 90 min at RT. Next, hybridization of the XMP Y orange probe (undiluted, Metasystems, ref: D-1421-050-OR) to the Y-chromosome was performed. After loading the probe into the hybridization chamber, those were sealed with rubber cement. To allow the probe to penetrate the tissue, samples were incubated for 3 h at 45 °C. Then, samples were transferred to a hot block at 80 °C for 5 min, to denature DNA. Afterwards, probes were hybridized with DNA for 2 days at 37 °C. Posthybridization consisted of $3 \times 15$ min washes with $2 \times$ SCC at 37 °C and $2 \times 5$ min stringency washes with $0.1 \times$ SCC at 60 °C. Finally, sections were counterstained with DAPI (1:15,000, Sigma).

**Quantification of fluorescent cells in retinal sections.** Fluorescent labelled cells, that is, eGFP, Ai9 (tdTomato) or Y-chromosome, were quantified on DAPI-stained serial sections of experimental retinas ($\geq 5$ biological replicates) using fluorescence microscopy (Apotome ImagerZ1, Zeiss). Respective data were plotted using GraphPad Prism version 6 (GraphPad Software).

**Flow cytometry.** Transplanted retinas ($n = 3$) and retinas from the used reporter strains, that is, Nrl-eGFP ($n = 3$) and DsRED ($n = 3$), were dissociated to a single-cell suspension using papain dissociation (Worthington Biochemical Corporation) for 20 min with 100 µg ml$^{-1}$ papain according to the manufacturer's instructions, followed by filtration through 30 µm cell strainers (Sysmex Partec). Samples were analysed for reporter fluorescence, that is, eGFP and DsRED, with a BD FACS-Canto II flow cytometer (BD Bioscience) after gating retinal cells for singlets (FSC-H versus FSC-A and SSC-H versus SSC-A) and photoreceptors. Flow cytometry data were processed offline using FlowJo software (Tree Star).

**Imaging flow cytometry.** For imaging flow cytometry, individual transplanted retinas ($n = 3$) were dissociated and processed as for conventional flow cytometry (see above). Control samples (that is, mixed eGFP and DsRED cells) were generated as follows: P4 Nrl-eGFP retinas and adult DsRED retinas were separately dissociated into a single-cell suspension and Nrl-eGFP rods were enriched using CD73-based magnetic-activated cell sorting in line with the procedure for transplantation (see above). Each individual DsRED retina ($n = 3$) was then mixed with $2 \times 10^5$ Nrl-eGFP rods and spun down for further processing. Transplantation and

control samples were analysed for reporter fluorescence (that is, eGFP and DsRED) using an ImageStream X Mark II imaging flow cytometer (Amnis), after gating retinal cells for singlets (aspect ratio versus area). For reasons of technical feasibility (limitation of data volume per data set per sample), the majority of mere DsRED-positive cells was gated out and hence excluded from data acquisition by using GFP fluorescence as a cutoff parameter. Data were processed offline using IDEAS software (Amnis). Acquired individual images of double-positive (GFP$^+$/DsRed$^+$) cells (73, 34 and 31 individual cells from transplanted retinas; 12, 5 and 4 individual cells from the mixed control samples) were analysed and quantified for the appearance of GFP and DsRED within the same cell, doublets and cells with attached fluorescent debris.

**Data availability.** The authors declare that all data supporting the findings of this study are available within the article and its Supplementary Information files or from the corresponding author upon reasonable request.

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

## Acknowledgements

We thank Sindy Böhme, Sabrina Richter and Katrin Sippel for animal husbandry, Drs Yun Z. Le and Mike O. Karl for providing B2-Cre mice, Drs Anand Swaroop and Sandra Cottet for providing Nrl-eGFP mice, and Drs Elly Tanaka, Gerd Kempermann, Mike O. Karl and Annette E. Rünker for advice and critically reading the manuscript. This work was supported by the Flow Cytometry Facility, a core facility of BIOTEC/CRTD at Technische Universität Dresden. This work was financially supported by the Deutsche Forschungsgemeinschaft (DFG) FZT 111 Center for Regenerative Therapies Dresden, Cluster of Excellence (M.A.), DFG Grant AD375/3-1 (M.A.) and the ProRetina Stiftung (M.A.).

## Author contributions

T.S.F., S.L. and O.B. contributed to the experimental design, data acquisition, data analysis and manuscript preparation. J.H. contributed to data acquisition. K.P. contributed to experimental design and data acquisition. M.A. contributed to experimental design, data analysis and manuscript preparation.

## Additional information

**Competing financial interests:** The authors declare no competing financial interests.

