## [Peer Review File · Nature Communications]

Reviewer #1 (Remarks to the Author)

The paper by Santos-Ferreira et al. titled "Retinal transplantation of photoreceptors results in donor-host cytoplasmic exchange" found a key problem with the interpretation of previous photoreceptor cell transplantation experiments to retina. In all previous experiments transplanted cells were labeled with a marker protein, and retinal integration was assessed by following the location and connectivity of cells marked with the marker protein. In a number of very elegant and clear experiments the authors demonstrate that marker proteins can move from donor cells to host cells and also from host cells to donor cells, making previous conclusions about integration of photoreceptors to the retinal circuit incorrect. [redacted] This is a very important paper that will cause a major reevaluation of results in the field of cell transplantation.

The paper is exceptionally well written and the experiments are exceptionally clever and performed in very high standard. I recommend publishing this paper immediately without any modifications. The authors should be congratulated to for the work.

Reviewer #2 (Remarks to the Author)

In their manuscript titled "Retinal transplantation of photoreceptors results in donor-host cytoplasmic exchange" Santos-Ferreira et al. describe a new mechanism of restored visual function from transplanted photoreceptor precursors. In the first experiment, they transplanted GFP+ cells into a RFP recipient. The presence of double positive cells suggested exchange of donor material with the recipient cells. Next, they used a donor expressing Cre-recombinase in photoreceptors and a donor with a reporter. The data from such transplantation suggested bi-directional exchange of material. As a complementary approach, they used EdU as an indelible marker of the transplanted cells. None of the recipient cells had EdU consistent with the conclusion that there was no fusion of nuclei. Y-chromosome FISH was also used to demonstrate that the nuclei did not fuse. [redacted]

Overall, the experiments are well done and the interpretation is appropriate. [redacted] However, there are some concerns about statistics and controls as follows:

- 1) It is not clear how many independent biological replicate experiments were performed for any of the experiments. For example, in Fig. 1, it says n=3. What does that mean? Were 1 million cells sorted from the same experiment 3 times? How many biological replicates were performed for this experiment and all experiments?

2) One control for the RFP-GFP experiment would be to dissociate a GFP+ donor at P4 and purify the cells. Then, mix those cells with dissociated RFP+ cells and spin down briefly while still viable. Then, resuspend and perform the flow sorting. It is possible that GFP+ debris is carried over and sticks to the RFP+ cells giving a false positive signal. This is just one example of the type of control that is missing from the study. The same could be done for Figure 2 experiments.

3) In general, biological replicates and false positive rate should be carefully documented to show statistical significance.

Reviewer #3 (Remarks to the Author)

There has been a great deal of excitement around the potential for photoreceptor transplantation in the past ten years. The results presented in this study challenge the interpretation of these transplantation studies, including some of those of the senior author. This study proposes that the cells transplanted to the subretinal space in mice do not migrate into host retinas, but instead "exchange intracellular proteins with host photoreceptors" resulting in the widespread misinterpretation of the previous studies. The report is organized into a series of different experiments, which all reach the same conclusion. One could quibble about the results of each one, but together they are convincing, and this report will dramatically change the field of retinal transplantation as much as the report in Nature of MacLaren et al in 2006.

Figure 1: The authors graft GFP donor cells into RFP host, they find double-labeled cells, suggesting fusion or exchange of the fluorescent proteins. These data are interesting, but not too convincing, since there could be contamination between cells during the enzymatic dissociation process. This could be a supplemental figure.

Figure 2. The authors next transplanted floxed-tdTomato rods to a B2-Cre host, and they found reporter positive cells within the retina and in the subretinal space. They conclude "This result demonstrates that Cre-recombinase produced by host photoreceptors translocates into donor cells enabling the excision of the stop cassette and thus expression of the reporter protein."

However, this is not conclusive, since it could be that the Cre travels from the host rods to the cells in the subretinal space, and then the donor rods pick it up prior to migrating into the host retina. The strong conclusion at this point should be qualified.

Figure 3. The authors then labeled donor mice with EdU as they were developing and looked for EdU+ rods in the host ONL. They found GFP+ cells in the ONL, but no EdU+ cells in the ONL, while "the majority" of the rods in the subretinal space were EdU+. However, there is

no quantitation, so it is possible that a specific subpopulation of cells is more migratory and this was generated on a day when the EdU was not given. I suggest they provide quantitation of the % labeled rods in the retinas of mice given the EdU injections at the intervals used in this experiment, and quantify the percentage of EdU+ rods in the subretinal space.

Figure 4. The authors next use a male Nrl-GFP mouse as the donor and transplant to a female recipient. Many GFP+ photoreceptors are present in the ONL, but only around 1% of these have the Y-chromosome. This is the most convincing experiment in the study, and it is hard to come to any other conclusion.

[redacted]

Dear Dr. LeGood,

Please find in the following our point-by-point responses to the reviewers' comments.

Reviewer 1:

The paper by Santos-Ferreira et al. titled "Retinal transplantation of photoreceptors results in donor-host cytoplasmic exchange" found a key problem with the interpretation of previous photoreceptor cell transplantation experiments to retina. In all previous experiments transplanted cells were labeled with a marker protein, and retinal integration was assessed by following the location and connectivity of cells marked with the marker protein. In a number of very elegant and clear experiments the authors demonstrate that marker proteins can move from donor cells to host cells and also from host cells to donor cells, making previous conclusions about integration of photoreceptors to the retinal circuit incorrect. [redacted] This is a very important paper that will cause a major reevaluation of results in the field of cell transplantation.

The paper is exceptionally well written and the experiments are exceptionally clever and performed in very high standard. I recommend publishing this paper immediately without any modifications. The authors should be congratulated to for the work.

Response:

We thank the Reviewer very much for this favorable and very positive judgement.

Reviewer 2

Comment 1:

Overall, the experiments are well done and the interpretation is appropriate.

Response:

We like to thank Reviewer 2 for the encouraging and positive view on our study.

Comment 2:

[redacted] However, there are some concerns about statistics and controls as follows:

Response:

We agree with Reviewer 2 that this part of our study has to be approached in more detail and therefore have deleted the data from the manuscript.

Comment 3:

It is not clear how many independent biological replicate experiments were performed for any of the experiments. For example, in Fig. 1, it says n=3. What does that mean? Were 1 million cells sorted from the same experiment 3 times? How many biological replicates were performed for this experiment and all experiments?

Response:

We are grateful about this comment to improve our manuscript and have addressed the Reviewer's concern regarding biological replicates. The number of biological replicates for each experiment are now mentioned in "Material and Methods" and in the corresponding figures. For the conventional and imaging flow cytometry analysis (Fig. 1) each 3 individual experimental retinas were used. Also the GFP+ and DsRED+ cell mixing control experiment was performed in triplicate, i.e. 3 independent experiments. In figures 2f and 3c (formerly figure 4c) each data point corresponds to a biological replicate (i.e. individual retina) as described in the "Material and Methods" section entitled: "Quantification of fluorescent cells in retinal sections".

Material and Methods section:

"Flow cytometry

Transplanted retinas (n=3) and retinas from the used reporter strains, i.e. Nrl-eGFP (n=3) and DsRED (n=3), were dissociated to a single cell suspension using papain dissociation (Worthington Biochemical Corporation)"

"Imaging flow cytometry:

For imaging flow cytometry individual transplanted retinas (n=3) were dissociated and processed as Individual DsRED retinas (n=3) were then each mixed with 2×10^5 Nrl-eGFP rods and spun down for further processing.”

Results section:

Figure 1: “.....(b) Flow cytometric analysis of individual experimental retinas (n=3) revealed a robust number of GFP⁺/DsRED⁺ double labelled photoreceptors (98 ±17 photoreceptors/1 million photoreceptors) within aDsRED host retinas three weeks after transplantation of eGFP⁺ donor cells (b, left plot),”

Figure 2: “.....(f) Quantification of tdTomato⁺ cells in the sub-retinal space and in the ONL of B2-Cre^{+/-} hosts (n=6) show similar amounts of reporter labelled cells (ONL: 443 ±103 cells/retina; SRS: 357 ±96 cells/retina). No tdTomato expressing cells were detected in transplanted B2-Cre^{-/-} mice (n=4).”

Figure 3: “(a) Sections of individual experimental retinas (n=5) were analysed by Y-chromosomal fluorescent *in situ* hybridization (Y-chromosome FISH) and immunohistochemistry for eGFP.”

Comment 4:

One control for the RFP-GFP experiment would be to dissociate a GFP⁺ donor at P4 and purify the cells. Then, mix those cells with dissociated RFP⁺ cells and spin down briefly while still viable. Then, resuspend and perform the flow sorting. It is possible that GFP⁺ debris is carried over and sticks to the RFP⁺ cells giving a false positive signal. This is just one example of the type of control that is missing from the study. The same could be done for Figure 2 experiments.

Response:

The Reviewer highlights a point that is important to strengthen our data and we appreciate the rigorous examination of our manuscript. We added the requested control experiment, i.e. mixing of enriched P4 GFP⁺ photoreceptors with DsRED⁺ retinal cells, besides repeating

the transplantation experiment and using imaging flow cytometry for a more thorough detection and evaluation of double fluorescent cells, since this flow cytometry technique allows the simultaneous visualization of individual cells. Indeed, as we now show in figure 1, all sorted cells positive for GFP and DsRED from the mixed control population represent doublets or cells with attached fluorescent debris (100%). In contrast, the majority (80%) of cells sorted positive for GFP and DsRED from transplanted retinas were truly double positive, i.e. showed GFP and DsRED fluorescence within the same cells. Concerning figure 2, we included a control experiment for the Cre/lox experiment in the supplemental data (see Supplementary Figure 2). Unfortunately, the suggested experiment, that is, mixing both Ai9 photoreceptors with B2-Cre^{+/-} retinal cells and analyses by flow cytometry would not yield an appropriate control for the Cre/lox transplantation experiment. The excision of the stop codon by the Cre-recombinase in the Ai9 photoreceptors and the onset of detectable tdTomato expression would take several days representing a too long period for sufficient survival of photoreceptors in vitro. Furthermore, the analysis of the Cre/lox experiment was performed on retinal sections rather than dissociated cells and thus would not be compatible to a flow cytometry control.

Comment 5:

In general, biological replicates and false positive rate should be carefully documented to show statistical significance.

Response:

We like to thank the Reviewer for his recommendation and have now added the missing data to the manuscript (see details in the responses to comments 3 and 4).

Reviewer 3:

Comment 1:

One could quibble about the results of each one, but together they are convincing, and this report will dramatically change the field of retinal transplantation as much as the report in Nature of MacLaren et al in 2006.

Response:

We like to thank Reviewer 3 for the positive feedback and her/his view about the importance of our finding.

Comment 2:

Figure 1: The authors graft GFP donor cells into RFP host, they find double-labeled cells, suggesting fusion or exchange of the fluorescent proteins. These data are interesting, but not too convincing, since there could be contamination between cells during the enzymatic dissociation process. This could be a supplemental figure.

Response:

We like to thank the Reviewer for raising this important issue. In line with the response to comment 4 of Reviewer 2, we have added further controls to the data set of figure 1 allowing the distinction between truly double positive cells and potential contamination via imaging flow cytometry. As the reviewer nicely pointed out, the combination of our results convincingly show that intracellular material is exchanged between donor and host photoreceptors. Therefore, we like to suggest to keep the modified figure 1 with the now added imaging flow cytometry analysis of transplanted retinas and corresponding cell mixing controls, as described in detail in the response to comment 4 of Reviewer 2, within the main part of the manuscript.

Comment 3:

Figure 2. The authors next transplanted floxed-tdTomato rods to a B2-Cre host, and they found reporter positive cells within the retina and in the subretinal space. They conclude "This result demonstrates that Cre-recombinase produced by host photoreceptors translocates into donor cells enabling the excision of the stop cassette and thus expression of the reporter protein."

However, this is not conclusive, since it could be that the Cre travels from the host rods to the cells in the subretinal space, and then the donor rods pick it up prior to migrating into the host retina. The strong conclusion at this point should be qualified.

Response:

The Reviewer raises an important point and we agree with the concern, as it was already stated in the manuscript “..... or, alternatively, could be due to integration of fluorescent labelled cells, following uni-directional transfer of Cre-recombinase, into the ONL.”. However, in combination with the other results presented in our study the possibility of tdTomato-positive cells migrating into the host retina is very rare (< 2%), but might still be happening, as shown in figure 3 (formerly figure 4), where few Nrl-GFP⁺/Y-chromosome⁺ were detected within the recipient ONL.

Comment 4:

Figure 3. The authors then labeled donor mice with EdU as they were developing and looked for EdU⁺ rods in the host ONL. They found GFP⁺ cells in the ONL, but no EdU⁺ cells in the ONL, while "the majority" of the rods in the subretinal space were EdU⁺. However, there is no quantitation, so it is possible that a specific subpopulation of cells is more migratory and this was generated on a day when the EdU was not given. I suggest they provide quantitation of the % labeled rods in the retinas of mice given the EdU injections at the intervals used in this experiment, and quantify the percentage of EdU⁺ rods in the subretinal space.

Response:

We agree with the comment of Reviewer 3 and indeed discussed similar experiments and quantitative analysis at the time we generated the EdU results. However, as we realized that 100% labeling of donor photoreceptors with EdU injections will not be possible, we decided to go forward and use the Y-chromosome approach, as documented in figure 3 (formerly Fig. 4), which resulted in adequate labeling of donor nuclei, thus allowing reliable quantification. As this experimental approach is also appreciated by the reviewer (see comment 5), we suggest to move figure 3 to the supplement (now Supplementary Fig. 3).

Comment 5:

Figure 4. The authors next use a male Nrl-GFP mouse as the donor and transplant to a female recipient. Many GFP+ photoreceptors are present in the ONL, but only around 1% of these have the Y-chromosome. This is the most convincing experiment in the study, and it is hard to come to any other conclusion.

Response:

We like to thank the Reviewer for this very nice comment.

Comment 6:

[redacted]

Response:

As already mentioned in our response to comment 2 of Reviewer 2, we agree that this part of our study has to be approached in more detail and therefore has been deleted from the manuscript.